# Long-term monitoring of margays (*Leopardus wiedii*): Implications for understanding low detection rates

**Bart J. Harmsen**[1,2]*, **Nicola Saville**[1,2], **Rebecca J. Foster**[1]

**1** Panthera, New York, New York, United States of America, **2** Environmental Research Institute, University of Belize, Belmopan, Belize

* bharmsen@panthera.org

**Data Availability Statement:** All relevant data are within the paper and its Supporting information files.

**Funding:** This study was supported by Panthera (https://www.panthera.org/) in the form of funds awarded to BJH and RJH, Wildlife Conservation

## Abstract

Population assessments of wide-ranging, cryptic, terrestrial mammals rely on camera trap surveys. While camera trapping is a powerful method of detecting presence, it is difficult distinguishing rarity from low detection rate. The margay (*Leopardus wiedii*) is an example of a species considered rare based on its low detection rates across its range. Although margays have a wide distribution, detection rates with camera traps are universally low; consequently, the species is listed as Near Threatened. Our 12-year camera trap study of margays in protected broadleaf forest in Belize suggests that while margays have low detection rate, they do not seem to be rare, rather that they are difficult to detect with camera traps. We detected a maximum of 187 individuals, all with few or no recaptures over the years (mean = 2.0 captures/individual ± SD 2.1), with two-thirds of individuals detected only once. The few individuals that were recaptured across years exhibited long tenures up to 9 years and were at least 10 years old at their final detection. We detected multiple individuals of both sexes at the same locations during the same survey, suggesting overlapping ranges with non-exclusive territories, providing further evidence of a high-density population. By studying the sparse annual datasets across multiple years, we found evidence of an abundant margay population in the forest of the Cockscomb Basin, which might have been deemed low density and rare, if studied in the short term. We encourage more long-term camera trap studies to assess population status of semi-arboreal carnivore species that have hitherto been considered rare based on low detection rates.

## Introduction

Camera traps are the standard tool for assessing population status of many terrestrial mammals, describing species distributions, and confirming the presence of some of the rarest terrestrial species (e.g. [1, 2]). Camera traps log detections of all species that trigger the sensors, including rare detections of previously unknown species (e.g. [2]). It has been assumed that species that overlap in their geographic ranges and are considered sympatric in some areas can be effectively detected under survey designs that do not necessarily take into consideration

Society (https://www.wcs.org/) in the form of funds awarded to BJH and RJH, Natural Environmental Research Council (UK) (https://nerc.ukri.org/) in the form of funds awarded to BJH and RJH, and Summerlee Foundation (https://summerlee.org/) in the form of funds awarded to BJH. The funders had no role in study design, data collection and analysis, decision to publish, or preparation of the manuscript.

**Competing interests:** No: The authors have declared that no competing interests exist.

behavioral micro-habitat preferences. Within terrestrial camera trap grids, detection rates vary between species depending on the spacing between camera traps relative to the species' home range size, and the micro-habitat of the camera traps relative to the species' natural history. If cameras are widely spaced compared to the average home range size, then there will be few spatial recaptures of individuals, and individuals existing only between camera traps will have zero probability of detection [3]. At the micro-habitat level, camera locations that are ideal for detecting one species may be sub-optimal for another species; for example, trail-based cameras favour species that walk trails versus those that prefer moving through dense undergrowth (e.g. [4]). Given the cost and effort of establishing and maintaining camera grids, researchers often use detections of non-target species to study little known animals, even though the camera survey design may be sub-optimal and the detection rates low for these 'by-catch' species (e.g. [5], clouded leopard (*Neofelis nebulosa*) [6], sun bears (*Helarctos malayanus*)).

Many solitary semi-arboreal carnivore species living throughout the Neotropics and tropical Africa and Asia are of unknown population status because they are difficult to sample [7]. Potentially, we could assess population status for these species by making use of the plethora of camera grids already deployed globally for surveying the larger, more easily-sampled terrestrial carnivores (e.g. jaguars (*Panthera onca*): [8], tigers (*Panthera tigris*) and leopards (*Panthera pardus*): [9]). Camera traps may be the most efficient method to sample their forays on the forest floor, however can they provide an unbiased representation of population status? If ground visits are infrequent or localized and cameras are spaced for optimizing detection of larger target species, detections of these smaller, semi-arboreal non-target species will be relatively rare. While rare detections indicate species presence in an area, we cannot assess population status unless we can determine whether low detection rate reflects true rarity of the species (low abundance in the area compared to sympatric species occupying a similar niche) or is an artifact of the sampling method (e.g. unsuitable camera spacing, sub-optimal camera trap locations). In this study, we assess population status of the margay (*Leopardus wiedii*), a neotropical, semi-arboreal felid, using data from a long-term camera trap study set up for monitoring the larger, sympatric jaguar (*Panthera onca*) in Belize, Central America [8].

Compared with other neotropical cats, the scientific community knows relatively little about margays [10]. They are associated with mature moist broadleaf forest, suggesting that they do well in protected forest interiors [11, 12]. Margays are well known for their climbing abilities, and evidence of arboreal hunting can be found in their diet [11]. There are no data on longevity in the wild, but margays have lived up to 20 years in captivity [11]. Compared with other felids in their guild, margays are small, weighing 2 to 4 kg, while jaguars weigh from 41 to 102 kg, and are largely confined to the forest floor [11]. We can distinguish individuals of both species from their individually unique pelt patterns, allowing assessments of abundance and distribution by tracking detections of individuals across camera locations [13, 14]. The geographic ranges of both species almost completely overlap, making margays potential by-catch in camera trap survey grids designed for jaguars [10, 15]. Jaguars have been widely studied using camera traps, with over 130 surveys run for estimating density [16–18]. There is great potential for extracting margay data from these numerous jaguar camera surveys and associated datasets. However, we expect that short-term surveys for jaguars are sub-optimal for sampling margays for population assessment, as the cameras are primarily deployed along trails, spaced widely compared to the margay home range [11], and under these sub-optimal sampling conditions are not run for long enough to amass sufficient captures and spatial recaptures to estimate population size [19]. Methods for assessing population status from camera data range from density estimation by mark-recapture analysis [12, 20], spatial distribution from occupancy modelling [21, 22], or at the simplest level, use of relative abundance indices (RAI) based on raw detections and trap effort [23, 24]. The more robust analyses, such as

mark-recapture or occupancy modelling tend to be data hungry and so may be unsuitable for use with sparse 'by-catch' data, leaving only RAIs for making weak inferences about the population.

Of 22 published camera trap studies that have documented margays, detection rates range from 0.04 to 2.64 detections per 100 trap-nights (mean +/- SD = 0.62 +/- 0.76, N = 30 estimates, across nine countries from northern Mexico to Argentina, S1 Data). Only four of the studies implemented surveys specifically designed to estimate margay density: three from a tropical rainforest site in Oaxaca, Mexico [25–27] and one from six sites in the Atlantic forest of southern Brazil [12]. Seven of the studies, across 13 sites, addressed ecological questions other than margay population status: geographic distribution [28]; occupancy [29]; habitat selection [30], felid coexistence [31]; activity patterns and/or relative abundance [14, 32, 33]; while 11 report on margay detections from camera trap surveys for which margays were not the target species [34–44]. For some large-scale camera surveys, the margays remain conspicuously undetected despite sampling in optimal habitat [31, 45–47]. We question whether low detection rates of margays throughout the camera trapping literature reflect species rarity or unsuitable sampling techniques. The problem of assessing rarity is that low sample sizes hinder robust estimation of density, abundance, occupancy and spatial distribution [21, 22, 48, 49]. In this study, we boost our sample size by using multiple seasons of margay by-catch obtained from 12 years of repeated surveys from the long-term jaguar-monitoring program [8]. While this long-term dataset is too sparse for robust mark-recapture analyses or single-season occupancy modelling, it is suitable for modelling multi-season occupancy, describing the spatial distribution of individuals through time, and analyzing the effect of trap effort on detections. In combination we use these approaches to distinguish between rarity and low detection rate associated with sub-optimal study design. Thus, we illustrate how to make use of by-catch data which may otherwise be used inappropriately or considered as too sparse for robust analysis and interpretation.

The application of occupancy modelling to camera trap data requires a large area to be sampled in order to ensure independence between camera sites while simultaneously sampling an adequate number of sites for parameter estimation. In most camera studies, especially for wide-ranging species, camera locations are generally limited in number and rarely independent. Furthermore, the interpretation of occupancy within camera trap studies is ambiguous, representing the probability of the species occupying an arbitrarily defined area around each camera location, which in reality can only detect species presence within an ~30m2 zone in front of the lens. We use this to our advantage, applying a multi-season occupancy model as a means to assess the methodological limitation of by-catch data from camera traps by estimating colonization and extinction probabilities as an index of the suitability of camera placement for margay detection. We assume that high colonization and extinction probabilities reflect inconsistent occupancy of camera locations by margays through time, implying sub-optimal camera placement. Conversely, low rates of colonization and extinction reflect consistent occupancy of camera locations from one year to the next, implying that cameras are well-placed for margay detection. In this way, multi-season occupancy modelling is useful for understanding margay presence in the survey grid and the suitability of the survey design for monitoring margays, however it does not provide clear insight on rarity. Where data are too sparse to estimate abundance directly, we can make use of patterns of the spatial distribution of individuals through time to further explore the question of rarity.

For a felid population in which individuals are rare, we expect that the spatial distribution of individuals will be characterized by a discontinuous distribution of large and, or, exclusive home ranges. Snap-shot short-term camera trap surveys with few capture records cannot provide sufficient data for the assessment of the spatial distribution of individuals. By combining

the data from repeat surveys across multiple seasons, we describe the distribution of individuals across the survey grid, estimate crude home range size from the spatial recaptures of individuals, and assess range overlap and interaction from the number of individuals detected at the same camera locations, and the time intervals between detections of different individuals at the same location [50]. In this way, we assess whether the ranging behavior reflects that of a rare carnivore (wide and/or discontinuous and/or exclusive ranges) or an abundant carnivore (small and/or continuous and/or overlapping ranges).

We further investigate whether the species is rare or has low detection rate associated with sub-optimal sampling design by looking at the effect of trap effort on detections, both the length of the survey period and the density of detectors. We assume that a species with low capture probability is rare if the number of recaptures increases but not the number of individuals, with increased sampling period and/or with increased density of detectors. Conversely, if we detect new individuals as we lengthen the sampling period within the limitation of a temporally closed population and/or increase detector density, we assume that the low capture probability was due to sub-optimal survey design rather than species rarity.

Because the study site has a homogeneous habitat structure, limited environmental fluctuations, and is free of anthropogenic disturbances, we assume stable population conditions across time [51]. Within this context, we assess (1) temporal variation in margay occupancy of the camera grid, (2) the influence of increasing trap effort on detecting margays, (3) distribution of individuals across the camera grid, (4) crude home range size, and (5) range overlap and interaction. To our knowledge this is the first long-term study of margays. In addition to assessing population status by understanding whether low detection rate is a methodological artefact or reflects true rarity, we also provide the first estimates of margay longevity in the wild, ranging behavior and territoriality. This study can guide the development of future studies of margay population structure and status.

## Methods

The study was conducted in the moist broad-leaved tropical forest of the Cockscomb Basin Wildlife Sanctuary (from hereon, CBWS or sanctuary), a natural area located in the south-central zone of Belize created to protect the forests, the fauna and the hydrographic basins of the eastern slopes of the Maya Mountains. The basin was selectively logged until 1981, and protected in 1986. The 490 km² sanctuary is now a mosaic of regenerating secondary forest in several stages of succession. Many of the old logging roads in the eastern part of CBWS are maintained as tourist trails or patrol routes (Fig 1, 16˚45'50.49"N / 88˚30'18.40"W; projection: GCS WSG84). A camera trapping grid originally established for long-term monitoring of jaguars has revealed relatively high densities of jaguars, pumas and ocelots within the CBWS when compared with the rest of Central America [8, 19, 50–52]. This study makes use of the photo records of margays. Permission for the study was granted by the government of Belize's Forest Department, and the on-site co-managing NGO, the Belize Audubon Society.

### Camera trapping

Twenty locations, with two cameras per station, were used annually to survey during 12 dry seasons (Feb-Jun) from 2003 to 2017 (Table 1). From 2003 to 2008, we used traditional film camera traps (CamTrakker, Cuddeback and DeerCam), with an enforced 3-minute delay between exposures to prevent excessive photos of herding species such as peccaries (*Tayassu sp.*). We were unable to deploy cameras in 2009 and 2010. From 2011 onwards, we used digital camera traps (Pantheracam) with a minimum delay of 8 seconds between successive photo triggers. The 20-station dry season survey duration ranged from 59 to 98 days per year

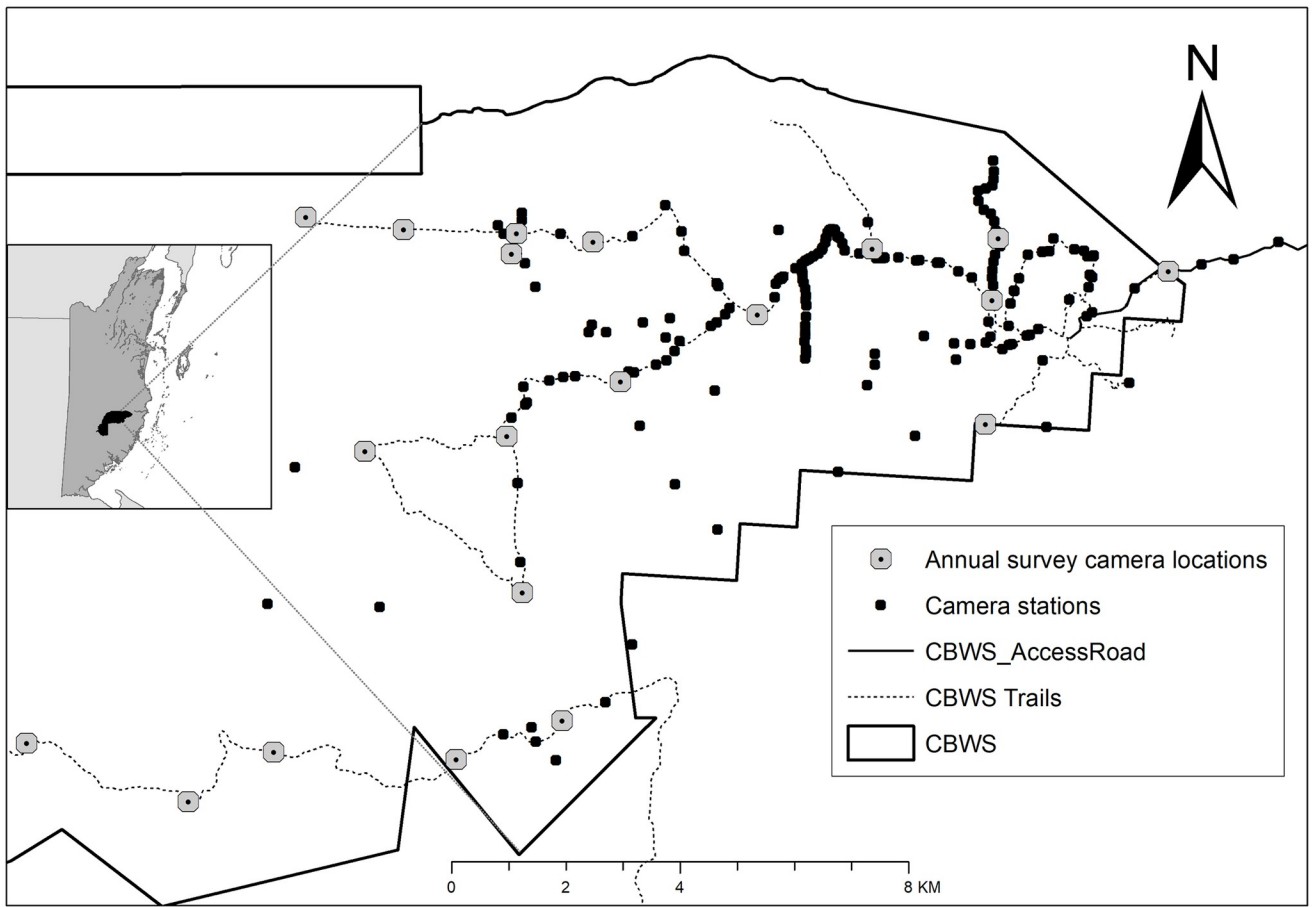

**Fig 1. The distribution of camera traps in the Cockscomb Basin Wildlife Sanctuary, Belize.** Black dots represent all camera stations deployed between 2003 and 2017, white dots represent the 20 locations surveyed annually for 12 dry seasons.

[8], with a continuous 365-day survey, spanning the 2013 and 2014 surveys ([19]; Table 1). We designed the camera distribution to optimize capture probability of jaguars, setting cameras at a height of approximately 40 cm above the ground, and the majority along the existing trail system [13, 53, 54]. From here-on we refer to the dry season survey as the 20-station survey. The 20-station survey grid covered a minimum convex polygon (MCP) of ~120 km² (~1.6 stations/10 km²). Neighbouring camera stations were on average 2.0 km apart (range 1.1 to 3.1 km) and the greatest distance between any two camera stations was 21.6 km. In addition to the 20-station survey, we ran additional small-scale surveys, which added to the total trap effort per year, and which could be analysed separately or in combination with the 20-station survey (Fig 1, Table 1; [4, 50, 54–56]). In particular, from May 2016 to June 2017, we deployed camera traps every 700 m along a 14 km of curving trail length within the annual survey grid as part of a study of marking behavior, and infrared video cameras (Browning Strike Force) deployed reactively to monitor scrape activity of cats (N = 55 stations, average distance between neighbouring stations was 0.14 km, and maximum distance between any two stations was 5.5 km, Table 1 [57], providing the highest density of cameras sustained for a year at this study site. Overall, we monitored 250 locations with camera traps between February 2003 and June 2017, with trap effort ranging from 5 to 2,111 functional trap-nights per station over the 14 years (Fig 1). We restricted most of our analyses to the

**Table 1. Camera trap survey information in the Cockscomb Basin Wildlife Sanctuary, Belize.**

| Year | Stations | Mean Dist (km) | Trap Effort (TN) | |
|---|---|---|---|---|
| | | (±SD) | Total | 20-station |
| 2003 | 33 | 1.17 (0.77) | 2,898 | 2,345 |
| 2004 | 84 | 0.80 (0.69) | 4,858 | 2,068 |
| 2005 | 30 | 1.98 (0.54) | 2,880 | 2,682 |
| 2007 | 31 | 1.77 (0.51) | 3,466 | 2,172 |
| 2008 | 50 | 1.29 (0.31) | 3,159 | 1,425 |
| 2011 | 19 | 1.98 (0.54) | 1,607 | 1,607 |
| 2012 | 20 | 1.98 (0.54) | 1,800 | 1,800 |
| 2013 | 30 | 1.22 (0.95) | 7,166 | 6,039 |
| 2014 | 23 | 1.57 (0.84) | 3,341 | 3,223 |
| 2015 | 77 | 0.50 (0.59) | 6,828 | 2,473 |
| 2016 | 72 | 0.46 (0.69) | 8,568 | 2,122 |
| 2017 | 48 | 0.67 (0.78) | 8,293 | 3,252 |
| **Mean (SD)** | **43 (22)** | **1.28 (0.55)** | **4,572 (2,391)** | **2,601 (1,170)** |
| '13–14 | 20 | 1.98 (0.54) | - | 9,262 |
| '16–17 | 55 | 0.14 (0.12) | 11,486 | - |

Number of camera trap stations deployed per year from 2003 to 2017 in the Cockscomb Basin Wildlife Sanctuary, Belize, showing the mean distance between neighboring stations and trap effort in trap-nights (TN) for all stations per year, the annual 20-station surveys, the year-long 20-station survey ('13–14) and the year-long high-density survey ('16–17)

subset of stations with $\geq$ 20 functional trap-nights (N = 236 locations). We performed all statistical analyses using R [58].

## Individual identification

Every camera trap photograph was stamped with the time and date. We identified individual margays based on their unique spot patterns, and assigned sex based on the presence or absence of testicles, as for jaguars, following [59]. We identified individuals by eye, and then used the pattern recognition software "Hotspotter" [60] to confirm all identifications. Because margays are small, and the camera traps were set at a height to detect larger felids, we frequently only detected one flank during a detection event [61]. Therefore, the number of individuals captured on the flank side with the most individuals would be the conservative estimate of number of detected individuals. We also calculated the maximum possible number of individuals detected, by summing left and right single flank individuals. To ensure temporal and spatial independence, we excluded repeat detections of the same individual at the same location on the same day.

## Activity pattern

Margays are considered nocturnal across their geographic range [11]. We tested the hypothesis that this nocturnal behaviour would be reflected in the 24h pattern of photo detections of margays on the forest floor. For multiple detections at the same location within the same hour, we used the median time event, although these occasions were rare. We used the R package 'overlap' to create a smoothed kernel density distribution of the daily time stamps of margay detections, separating events into detections of males and females, excluding those for which sex could not be assigned [62, 63]. We estimated the level of temporal overlap between the sexes using the coefficient of overlap (Dha) between the two populations [62, 63]. A Dha value of 0

indicates no overlap, and a value of 1 indicates identical activity patterns. We estimated the precision of the coefficient of overlap by bootstrapping with 999 simulations [62].

**Occupancy and variation between years.** We investigated the pattern of margay detections and non-detections across the years as an index of the suitability of the annual 20-station survey for detecting margays. For this we used a multi-season occupancy modelling framework using the colext function in the r-package 'unmarked' [64] to estimate detectability ($p$, the probability of detecting a margay if present), colonisation ($\gamma$, the probability of an unoccupied camera location being occupied the following year) and extinction ($\varepsilon$, the probability of an occupied camera location being unoccupied the following year). Within this framework, occupancy ($\psi$, the proportion of camera locations occupied by a margay) is estimated for the first season only (year 1) and thereafter derived from the other three parameters. We used the first 90 days of each of the 12 annual 20-station surveys and divided each into three secondary sampling occasions of 30 days. For each secondary sampling occasion, we recorded the detection history (margay detection or non-detection) at each of the 20 camera locations. The mean length of the primary period between each 90-day survey was 1.3 years (range 1 to 3 years, n = 11 primary periods for 12 surveys).

We considered all combinations of models in which $p$, $\gamma$ and $\varepsilon$ were either constant or varied across primary periods (i.e. between years) and in which $p$ was either constant or varied across secondary periods (i.e. between occasions within the same year). We ranked the models using Akaike's Information Criterion corrected for small sample size (AICc) [65, 66]. We tested for goodness-of-fit of models following [67], using the r package 'AICcmodavg' [68]. We used 1,000 bootstrap simulations and estimated the over-dispersion parameter $\hat{c}$ [68]. We used $\hat{c}$ from the most parameterized model as a variance inflation factor to correct for overdispersion and adjust our model selection [67, 68]. We present the resulting Quasi Akaike's Information Criterion corrected for small sample size (QAICc) and assume substantial support for the models for which $\Delta$QAICc is $< 2$ relative to the minimum QAICc.

We interpret the model estimates of $\gamma$ and $\varepsilon$ within the context of understanding how well-placed are the camera traps for detecting margays. We assume that a high probability of camera locations switching from being unoccupied by a margay to occupied ($\gamma$), or vice versus ($\varepsilon$), from one year to the next, reflects inconsistent occupancy of camera locations by margays through time, implying sub-optimal camera placement, especially if $\gamma$ and $\varepsilon$ fluctuate widely between primary periods. We considered $\gamma$ and $\varepsilon$ to be high if $> 0.1$, equivalent occupancy of two stations in a previously unoccupied grid (or the loss of margays from two stations in a 20-station camera grid that had been fully occupied).

**Influence of trap effort on margay detections.** We investigated the influence of changing temporal and spatial trap effort on the margay detection rate, specifically margay presence (detection or non-detection of margays), number of margay detection events (captures plus recaptures), number of margay individuals, and number of spatial recaptures per individual, if this exceeded 1 location. We used the entire dataset to estimate the minimum sampling period per station necessary to detect margay activity on the forest floor, and the average margay detection rate. We tested for a linear relationship between the number of margay detection events and trap effort per station across the full 12-year period.

Using a Welch's T-test, we compared the trap effort per year between stations where we detected margays versus those where we did not detect margays. The unit of analysis was the pooled data from a camera station location for one calendar year, hereafter referred to as a 'location-year' (see [50]). Thus, the same station could have location-year in the categories of "margay detection" and "margay non-detection".

We assumed that if margays are abundant in the study area but have low capture probability due to sub-optimal design of the 20-station survey grid, then we would detect new individuals

with increasing sampling period. Conversely, if margays are rare in the study area, we would detect no new individuals despite repeated detections (recaptures) of the same few individuals with increasing sampling period. Therefore, we tested for linear relationships between the number of individuals detected and (a) the trap effort per survey, and (b) the number of margay detection events per survey.

We assumed that if camera stations are optimally spaced for margay detection, then the proportion of individuals with spatial recaptures (detections of the same individual at >1 location) would increase with sampling period, indicative that margay home range spans multiple locations. However, if the stations are too widely-spaced, then the proportion of individuals with spatial recaptures, would not increase with sampling period, because individuals do not travel far enough to reach more than one camera location. We investigated this by comparing the proportion of individuals with 1, 2, 3,... *n* detections (recaptures) and the proportion of individuals detected at 1, 2, 3, ... *n* stations (spatial recaptures) between four datasets differing in trap effort (sampling period and/or detector density). As a baseline, we used the 11 20-station surveys (i.e. excluding 2013–2014 which ran for one year), calculating the proportion of individuals with recaptures and spatial recaptures within survey years. We assessed the influence of increased sampling period using the data from the same 20-station survey grid run for a year (2013–2014), and across multiple years using pooled data across all 12 20-station surveys; and the influence of increased camera density using the high-density survey run within the same survey grid for a year (2016–2017).

**Ranging behavior.** For margays with spatial recaptures, we assessed ranging behavior and quantified range size by calculating the maximum distance moved (MDM) by each individual between camera traps. By using data at three temporal and spatial scales, we calculated MDM per individual at different levels of effort as a proxy of home range size, and to detect large-scale moves (dispersals or shifts in home range) within and between years:

1. MDM over 365 days, using the high-density camera survey (2016–2017) for which the maximum distance between stations was 5.5 km, and the mean distance between neighboring stations was 0.14 km.

2. MDM per calendar year (2003 to 2017), using data from all camera locations within each year. Each calendar year included an annual 20-station survey for which the maximum distance between stations was 21.6 km, and the mean distance between neighbouring stations was 1.28 km), so allowing for detections of long-range movement within any given year.

3. MDM over 14 years (12 surveys), using all camera locations across all years thus allowing for detection of movement between years. Maximum distance between stations was 21.6 km, and the mean distance between neighbouring stations was 1.28 km

For each of the three datasets, we used the core r package [58], to calculate a kernel density distribution of the maximum distance moved (MDM) by margay individuals. We compared the three distributions to assess how estimates of MDM are influenced by the spatial and temporal scale of the camera surveys. We calculated crude estimates of home range size, using $\pi$ (½MDM)$^2$, assuming circular ranges for which MDM represents the diameter.

**Overlap.** We investigated the level of exclusive home range use, by calculating the rate at which we detected >1 individual at the same location. We used a static interaction framework, quantifying the number of individuals detected at the same location within any given year, and a dynamic interaction framework, investigating time difference between detections of different individuals at the same location. We assume that with an increase in the number of individuals of the same sex at the same location, the higher the deviation from exclusive home range use.

### Static interaction

The unit of analysis was a 'location-year'. The average sampling period per location-year was 140 (SD± 77) days, ranging from 18 to 365 days. Stations deployed across multiple years, contributed multiple location-years. We quantified the number of location-years that detected 1, 2, 3 . . . *n* individuals. For the subset of location-years with multiple individuals for which we could determine sex, we quantified the number of location-years that detected male-male, female-female and male-female dyads from the known set of sexed individuals.

### Dynamic interaction

For the subset of location-years with multiple individuals, we calculated the time difference between the detections of two individuals (hereafter referred to as 'dyads'). We used kernel density distributions, using the core r package [58], to describe the distribution of time intervals between detections of pairs of individuals constituting the dyads. We assessed this separately for male-female dyads (FM), male-male dyads (MM), and female-female dyads (FF). As a conservative measure of interaction, we assumed that individuals detected > 28 days apart from each other were independent events (no interaction). We based this on studies of decay rates of markings and tracks of jaguars and pumas in the same study area [57].

**Spatial distribution of margay detections.** To assess whether the distribution of margay detections across the study area was patchy, uniform or random, we plotted the camera locations with margay detections against those without detections and assessed by eye.

**Survival.** For all individuals with multiple detections, we calculated the number of years between first and last detection as an estimate of minimum longevity of margays in the study area. Margays reach sexual maturity between 12–18 months [11], therefore we estimated the minimum age plus 1 year, assuming that the individual was at least one year old when first detected.

## Results

We detected 411 independent margay events over 54,864 trap-nights across 236 locations and 12 years. Most detections occurred at night, showing overlap between the activity patterns of males and females (Dha value = 0.81; CI: 0.70–0.90 Fig 2). Males were more active than females during the late afternoon and early evening, with a peak in activity at 02:00. Female activity peaked at 03:00, and they remained more active than males during the latter part of the night and early morning until 07:00. Activity of both sexes remained low between 09:00 and 14:00, with a small but noticeable peak in female activity at 13:00.

### Occupancy and variation between years

For the 12 annually repeated 20-station surveys, we detected 252 margay events from 31,208 trap-nights. Detections rates of margays were variable and generally low for all the annual surveys, ranging from 0.14 to 1.5 detections per 100 trap-nights, and on average we detected 14 margay individuals per year (range 3–34; Table 2). The number of camera locations where margays were detected varied between the 12 20-station surveys from 2 to 13 locations (mean = 7 ±SD 3). However, over the entire period (12 annual surveys, 2003 to 2017) margays were detected 19 of the 20 camera locations.

We found greatest support for the occupancy model for which colonisation ($\gamma$) and extinction ($\varepsilon$) varied between primary periods (annual surveys) but detectability was constant (*p*; Table 3). For this model, $\gamma$ varied from 0.00 to 0.90 (mean = 0.44 ± 0.32) and $\varepsilon$ varied from 0.00 and 0.78 (mean = 0.40 ±0.23; Table 4). The initial occupancy, $\psi$, was 0.52 (± 0.15)

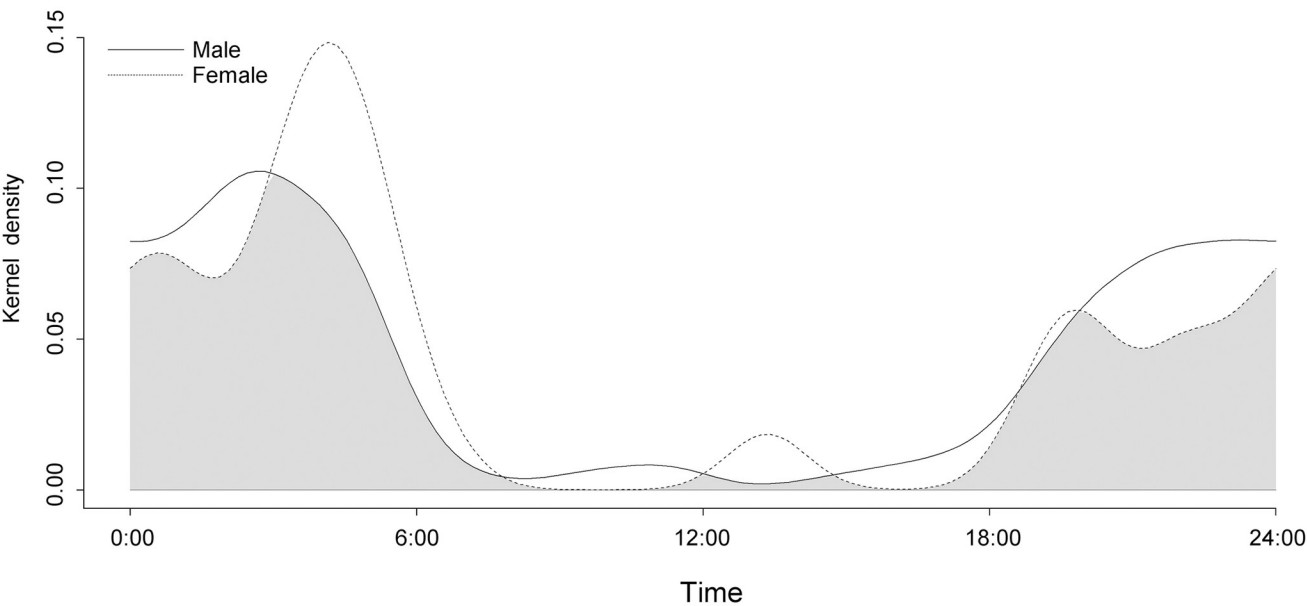

**Fig 2. Activity distribution of male and female margays in the Cockscomb Basin Wildlife Sanctuary, Belize.** 24h activity distribution for males and females, with activity overlap in grey shading, N = 179 male and 76 female independent photo events.

suggesting that half of the sites were occupied by margays, with occupancy derived for subsequent years ranging widely from 0.16 to 0.89. Detectability was low with a probability of 0.34 of detecting a margay at an occupied site over 30-days (Table 4). Fit of this model was adequate (Chi-Sq = 87.38, p > 0.1, Table 4) and over-dispersion of the observed data was minimal

**Table 2. Camera trap detections of margays in the Cockscomb Basin Wildlife Sanctuary, Belize.**

| Year | All stations | | | | | | 20-station survey | | | | | |
|---|---|---|---|---|---|---|---|---|---|---|---|---|
| | Male | Female | Unk | Total | Det | Det/100TN | Male | Female | Unk | Total | Det | Det/100TN |
| 2003 | 3–4 | 3 | 2–4 | 8–11 | 19 | 0.66 | 2 | 2 | 2–3 | 6–7 | 12 | 0.51 |
| 2004 | 8–9 | 3 | 5–10 | 16–22 | 30 | 0.58 | 6–7 | 2 | 2–5 | 10–14 | 18 | 0.87 |
| 2005 | 2–3 | 1 | 1 | 4–5 | 12 | 0.42 | 2–3 | 1 | 1 | 4–5 | 12 | 0.45 |
| 2007 | 9–10 | 5 | 6–8 | 20–23 | 36 | 1.04 | 9–10 | 4 | 6–7 | 19–21 | 31 | 1.43 |
| 2008 | 10–11 | 4 | 2–3 | 16–18 | 29 | 0.89 | 7 | 3 | 1–2 | 11–12 | 21 | 1.47 |
| 2011 | 5 | 0 | 5 | 10 | 13 | 0.81 | 5 | 0 | 5 | 10 | 13 | 0.81 |
| 2012 | 3 | 0 | 0–1 | 3–4 | 8 | 0.44 | 3 | 0 | 0–1 | 3–4 | 8 | 0.44 |
| 2013 | 14–16 | 10–11 | 4–7 | 28–34 | 55 | 0.77 | 14–16 | 10–11 | 4–7 | 28–34 | 55 | 0.91 |
| 2014 | 8 | 7–8 | 3–7 | 18–23 | 36 | 1.08 | 8 | 7–8 | 3–7 | 18–23 | 36 | 1.12 |
| 2015 | 10–11 | 3 | 9–13 | 22–27 | 38 | 0.56 | 7 | 1 | 6–8 | 14–16 | 19 | 0.77 |
| 2016 | 6–8 | 5 | 8–13 | 19–26 | 57 | 0.67 | 0 | 0 | 3 | 3 | 3 | 0.14 |
| 2017 | 11–12 | 6 | 6–17 | 23–35 | 78 | 0.94 | 7 | 2 | 4–10 | 13–19 | 24 | 0.74 |
| **Mean (SD)** | | | | | **34 (21)** | **0.74 (0.22)** | | | | | **21 (14)** | **0.80 (0.40)** |
| '13–14 | - | - | - | - | - | - | 15–17 | 11–13 | 8–13 | 34–43 | 87 | 0.94 |
| '16–17 | 7–12 | 7 | 8–14 | 22–33 | 97(12) | 0.95 | - | - | - | - | - | - |

Margay detections by camera trap stations deployed from 2003 to 2017 in the Cockscomb Basin Wildlife Sanctuary, Belize, showing number of margay individuals (males, females and unknown sex, unk), number of independent margay detection events, and detection rate (margay detections per 100 trap-nights) for all stations, the annual 20-station surveys, the year-long 20-station survey ('13–14) and the year-long high-density survey ('16–17)

**Table 3. Model selection for multi-season occupancy analysis of margays in the Cockscomb Basin Wildlife Sanctuary, Belize.**

| Models | nPars | QAICc | delta Δ | QAICcwt | cum. Wt | Goodness of fit | *p*-value | $\hat{c}$ |
|---|---|---|---|---|---|---|---|---|
| 1) ψ0(.)γ(t)ε(t)p(.) | 25 | 335.32 | 0 | 1 | 1 | 87.38 | 0.35 | 1.05 |
| 2) ψ0(.)γ(t)ε(t)p(t*s) | 60 | 359.59 | 24.27 | 0 | 1 | 46.80 | 0.17 | 1.25 |
| 3) ψ0(.)γ(.)ε(.)p(t*s) | 40 | 365.50 | 30.18 | 0 | 1 | 63.73 | 0.07 | 1.41 |
| 4) ψ0(.)γ(t)ε(t)p(t) | 36 | 370.38 | 35.06 | 0 | 1 | 75.38 | 0.34 | 1.05 |
| 5) ψ0(.)γ(.)ε(.)p(.) | 5 | 489.23 | 153.92 | 0 | 1 | 109.81 | 0.12 | 1.17 |
| 6) ψ0(.)γ(.)ε(.)p(s) | 7 | 495.64 | 160.32 | 0 | 1 | 102.21 | 0.22 | 1.10 |
| 7) ψ0(.)γ(.)ε(.)p(t) | 16 | 1030.57 | 695.26 | 0 | 1 | 96.40 | 0.09 | 1.25 |

Seven models ranked according to Quasi Akaike Information Criterion corrected for small samples sizes (QAICc) with parameters (ψ = occupancy, γ = colonisation, ε = extinction, *p* = detection probability) held either constant through time, "(.)"or allowed to vary between surveys, "(t)", and/or between secondary occasions (30 days),"(s)", nPars = number of parameters in model, QAICc = total QAICc score, delta Δ = the relative difference in QAICc values from the model with the smallest QAIC value, QAICcwt = weighted QAICc, cum. WT = accumulative weighted QAICc, Goodness of fit = total Chi Square, *p*-value (poor fit assumed for models where p < 0.1), $\hat{c}$ = over-dispersion parameter, where $\hat{c} > 1.0$ indicates over-dispersion and <1.0 indicates under-dispersion

compared to the model ($\hat{c}$ = 1.05). The next best ranking model had limited support and relatively poor fit to the data (Table 3).

## Detection of individuals and sex ratio

We detected margays at 87 of the 236 camera locations over 12 years (mean trap effort 220 ± SD 416 trap-nights per location, N = 236 locations). Of the 408 independent detection events, we were able to assign 371 (93%) to unique individuals. We identified at least 125 margay individuals (maximum of 187). Detection rate per individual was generally low (Table 2). For the subset of individuals for which sex could be assigned, we recorded 114 detections of at least 55 male margays (maximum 73 males) and 55 detections of at least 28 female margays (maximum 36 females) indicating a male to female detection ratio of 2:1, both in terms of number of individuals and number of detections. We found no difference between the sexes in the number of detections per individual (males mean ±SD = 2.5 ± 2.5 detections per

**Table 4. Estimated model parameter values for multi-season occupancy model of margays in the Cockscomb Basin Wildlife Sanctuary, Belize.**

| Model | Survey | ψ (±SE) | γ (±SE) | ε (±SE) | p (±SE) |
|---|---|---|---|---|---|
| ψ0(.)γ(t)ε(t)p(.) | 1 | 0.52(±0.15) | 0.80(±0.30) | 0.78(±0.20) | 0.34(±0.04) |
| | 2 | 0.50(±0.23) | 0.00(±0.00) | 0.67(±0.22) | 0.34(±0.04) |
| | 3 | 0.16(±0.25) | 0.90(±0.20) | 0.39(±0.49) | 0.34(±0.04) |
| | 4 | 0.86(±0.16) | 0.46(±1.01) | 0.49(±0.21) | 0.34(±0.04) |
| | 5 | 0.51(±0.17) | 0.54(±0.31) | 0.30(±0.32) | 0.34(±0.04) |
| | 6 | 0.62(±0.22) | 0.52(±0.33) | 0.44(±0.25) | 0.34(±0.04) |
| | 7 | 0.55(±0.20) | 0.10(±0.27) | 0.27(±0.25) | 0.34(±0.04) |
| | 8 | 0.44(±0.18) | 0.81(±0.22) | 0.00(±0.00) | 0.34(±0.04) |
| | 9 | 0.89(±0.13) | 0.00(±0.22) | 0.37(±0.19) | 0.34(±0.04) |
| | 10 | 0.56(±0.16) | 0.10(±0.19) | 0.64(±0.23) | 0.34(±0.04) |
| | 11 | 0.24(±0.24) | 0.62(±0.21) | 0.07(±0.38) | 0.34(±0.04) |
| | 12 | 0.70(±0.17) | x | x | 0.34(±0.04) |

Estimates of occupancy (ψ), colonisation (γ), extinction (ε), and detection probability (p) for the best fitting model (see Table 3) of multi-season occupancy of margays from 12 20-station annual surveys conducted 2003 to 2017.

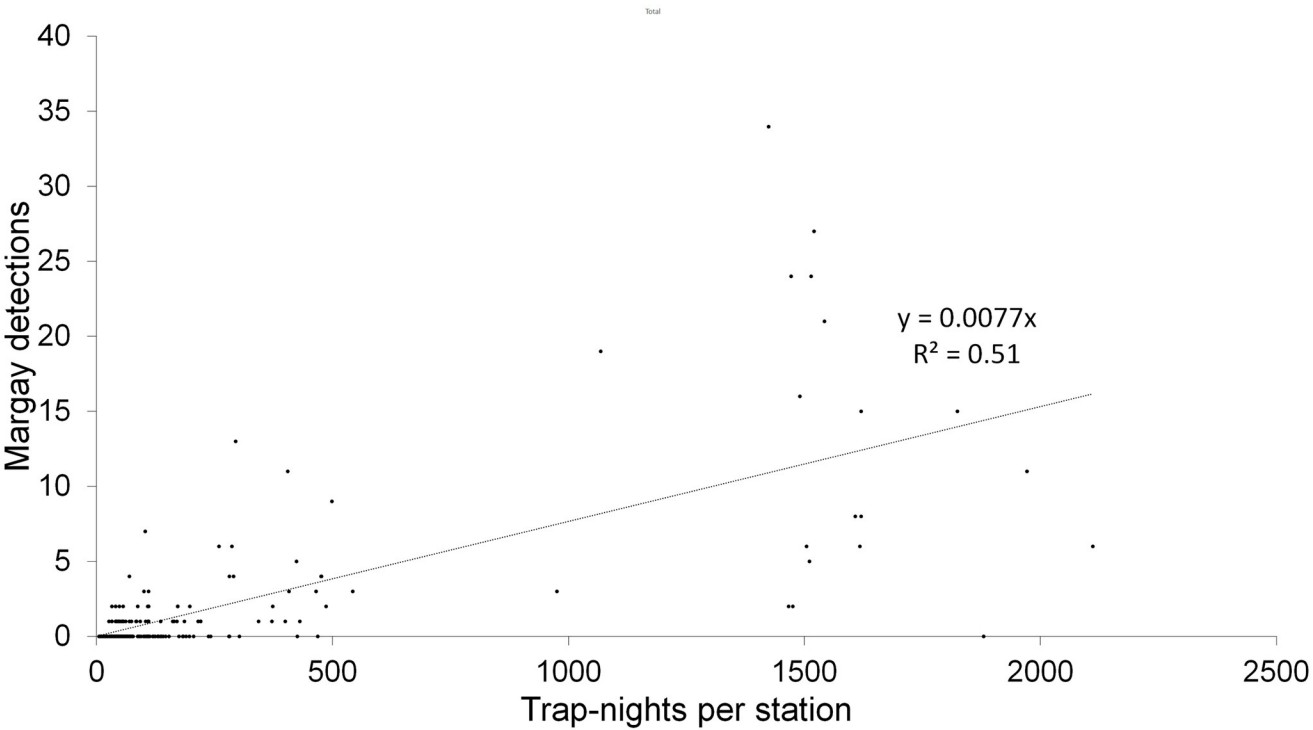

**Fig 3. Relation between margay detections and trap effort in the Cockscomb Basin Wildlife Sanctuary, Belize.** Variation in the number of margay detections per camera location with trap effort (functional trap-nights) for camera traps deployed, from 2003 to 2017(N = 250 camera locations).

individual, n = 73; females: 2.1±1.7, n = 36; Welch T-test on log transformed data for normality: t value = 0.79, df = 75.7, $p > 0.1$).

## Influence of trap effort on margay detections

Using the total trap effort across the 12 years, the number of detections of margays increased with trap effort, with a mean rate of approximately one detection of a margay every 100 trap-nights per station (y = 0.008x, $p < 0.001$ $R^2$ = 0.51, N = 250 stations, Fig 3). Trap effort was lower at location-years where we did not detect margays compared to location-years where we did detect margays (mean trap effort +/- SD: with margay detections 141 ± 77 trap-nights per location-year, n = 184 location-years; without margay detections: 87 ± 64 trap-nights per location-year, n = 333 location-years; Welch T-test on log transformed data for normality: t value = -9.29, df = 479.4, $p < 0.001$). Margays were more likely to be detected at stations that functioned for a least 100 trap-nights: we detected margays at 60 of 94 stations (69%) that functioned for ≥100 trap-nights, and at only 27 out of 129 stations (17%) that functioned for <100 trap-nights (Chi-Sq = 56.0, df = 1, p < 0.0001). The number of individuals detected per survey increased with the trap effort of the 12 annual 20-station surveys at a rate of approximately 4 individuals per 1,000 trap-nights per survey (individuals per survey = 0.0044 (trap-nights per survey), $p < 0.01$, $R^2$ = 0.56, N = 12 surveys, Fig 4). However, the recapture rate of individuals was generally low. The number of individuals increased with the number of detections at a rate of one new individual for every two margay detections per survey (individuals per survey = 0.53 (detection events per survey), $p < 0.0001$, $R^2$ = 0.94, N = 12 surveys, Fig 5), illustrating that detections were generally of new individuals rather than recaptures of known

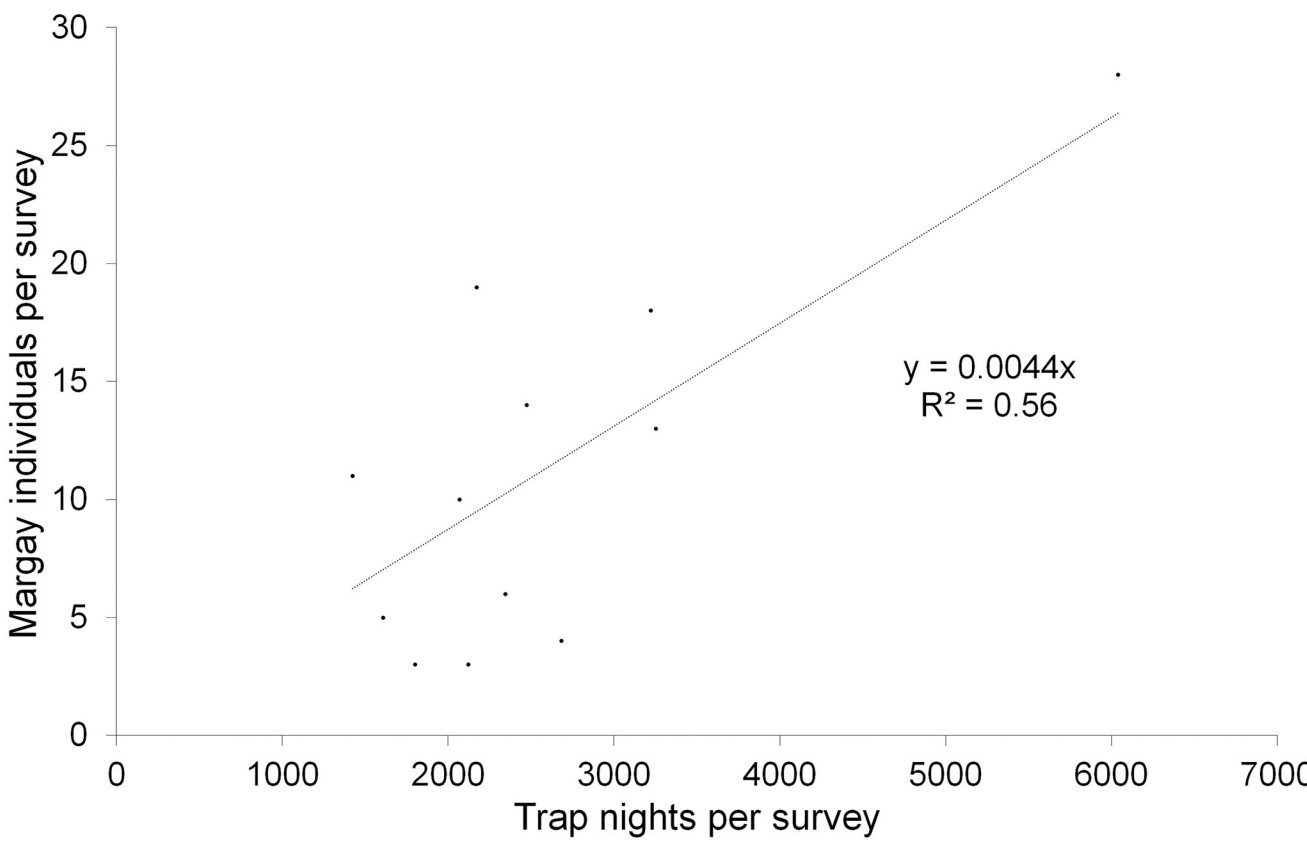

**Fig 4. Relation between number of margay individuals and trap effort in the Cockscomb Basin Wildlife Sanctuary, Belize.** Variation in the number of margay individuals detected per 20-station annual survey with trap effort (number of functional trap-nights, N = 12 annual 20-station surveys).

individuals. Overall, we recaptured only one-third (64) of the 187 individually identified margays.

The proportion of individuals that we recaptured increased minimally with increased sampling period (Fig 6a, 6b and 6c; black bars). During the 20-station surveys (2–3 months), we recaptured only 15% of individuals within the same survey (Fig 6a). When we extended the sampling period of the same survey to one year, we recaptured 42% of individuals (Fig 6b). When we pooled the data for the twelve 20-station surveys, allowing recaptures across years (2003 to 2017), we recaptured approximately one-third (31%) of individuals (Fig 6c). Spatial recaptures (detections at >1 station) were rare within the 20-station survey grid, regardless of temporal effort (Fig 6a, 6b and 6c; grey bars). Spatial recaptures were more common when we reduced the trap spacing by increasing the density of camera locations. Reducing the average trap spacing to < 1 km, increased the proportion of individuals with multiple captures and with detections at multiple stations (Fig 6d).

## Ranging behavior

Of the 187 individually identified margays, we detected only 32 (17%) at more than one location. The maximum distance between spatial recaptures was generally low, increasing marginally with increased spatial and temporal trap effort (detector density and trap-nights). For the high-density survey conducted over one year, with a maximum distance between

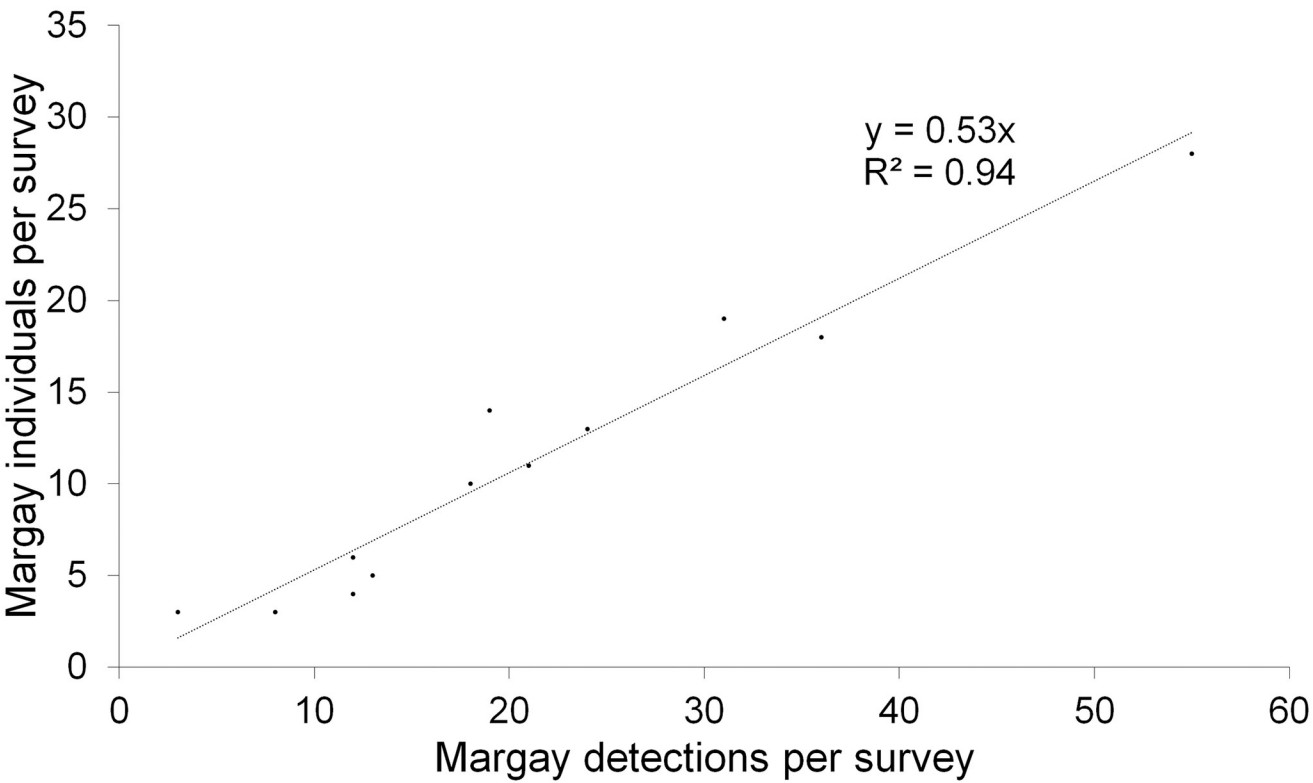

**Fig 5. Relation between number of margay individuals and detections in the Cockscomb Basin Wildlife Sanctuary, Belize.** Variation in the number of margay individuals detected per 20-station survey with the number of margay detections (N = 12 annual 20-station surveys).

cameras of 5.5 km and an average spacing of 0.14 km (Table 1), the majority of margays with spatial recaptures did not move further than 2 km/year (median MDM = 0.7 km, maximum = 2.1 km, N = 13 individuals; Fig 7, thick line). Likewise, when considering surveys on a larger spatial scale which had the potential to detect moves beyond 5.5 km/year (maximum distance between stations = 21.6 km, with an average spacing of ~ 1.28 km/year, ranging from 0.14 to 1.98 km/year for 12 years, Table 1), the majority of individuals did not move further than 2 km/year during any given year (median MDM = 1.2 km, maximum = 3.1 km, N = 25 individuals; Fig 7, dashed line). When pooling these data to increase the temporal effort and allow movement across years from 2003 until 2017, the MDMs increased marginally (median = 1.6 km, maximum = 6 km, N = 32 individuals; Fig 7, thin line). Detection of long-distance moves (dispersal events or range shifts) were rare. The multi-year MDMs of only four individuals exceeded the maximum MDM detected within one year, ranging from 3.2 to 6.0 km. The furthest move (6.0 km) was by a male that we detected only twice, first in 2013 and again in 2017.

For the subset of individuals that we identified as males or females, using all camera locations from 2003 to 2017, males ranged further than females, (mean MDM ± SD: males = 1.8 ±1.4 km, n = 16; females = 0.5 ±0.3 km, n = 5; Welch T-test on log transformed data for normality: t value = 3.0, df = 7.0, $p < 0.05$). Using MDM as a proxy for home range diameter, and assuming a circular home range, ranges of males were approximately 10x the size of female ranges (male = 2.5 km$^2$; female = 0.2 km$^2$). The MDMs of 2, 3, 4, and 6 km approximate to circular ranges of 3.1, 7.1, 12.6, and 28.3 km$^2$, respectively.

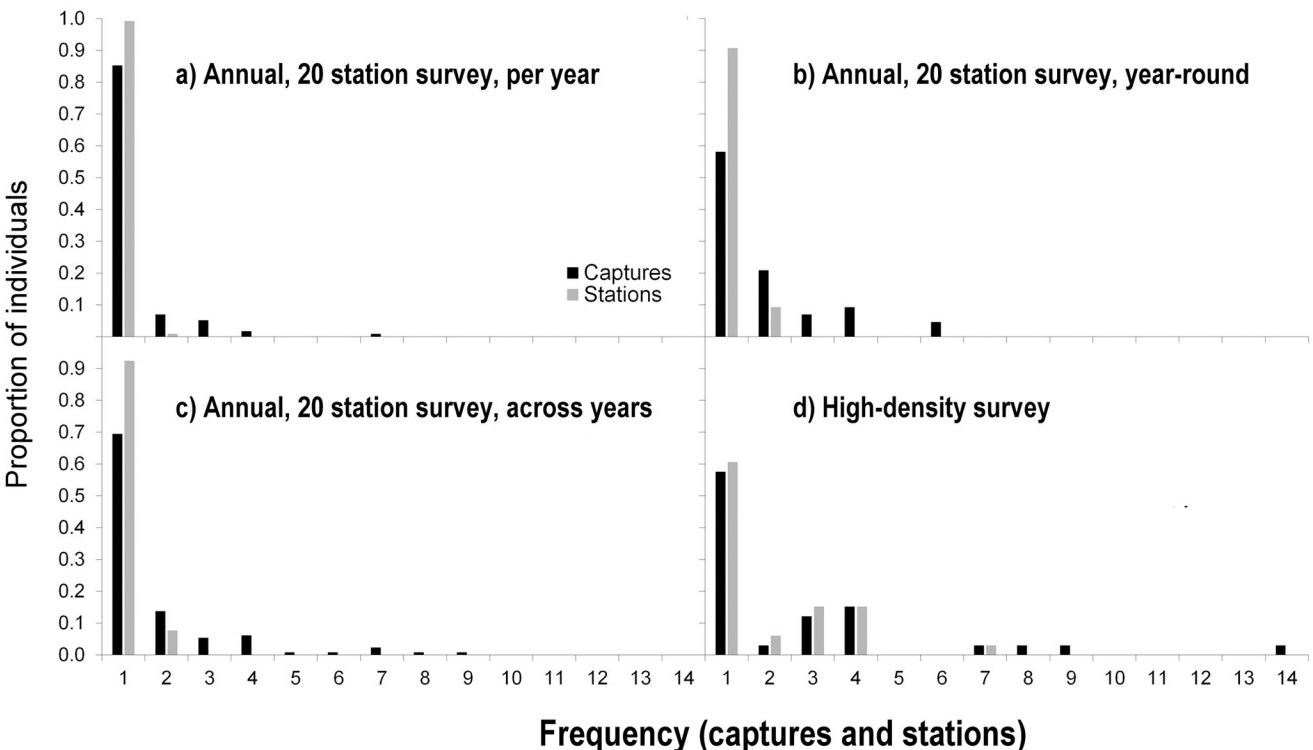

**Fig 6. Variation in number of individuals and (spatial) detections across various survey types in the Cockscomb Basin Wildlife Sanctuary, Belize.** The proportion of margay individuals detected with 1, 2, 3, . . .15 captures (black bars), and proportion of margay individuals detected at 1, 2, 3 . . .x stations (spatial recaptures, grey bars) for surveys differing in temporal and spatial effort: **a)** 12 annually repeated dry season 20-station surveys (2–3 months/survey) assessing detections per year (N = 115 individuals, mean distance between neighbouring stations = 2.0 km, mean trap effort ±SD = 2,195 ± 507 trap-nights/survey); **b)** Continuous year-round (2013–2014) 20-station survey (N = 43 individuals, mean distance between neighbouring stations = 2.0 km, trap effort = 9,262 trap-nights); **c)** 12 annually repeated 20-station surveys assessing detections across years (N = 131 individuals, mean distance between neighbouring stations = 2.0 km, total effort = 31,208 trap-nights); **d)** Continuous year-round (2016–2017) high-density survey (N = 33 individuals, mean distance between neighbouring stations = 0.14 km, trap effort = 11,486 trap-nights).

### Overlap and interaction

**Static interaction.**   Of the 175 "location-years" (pooled data for a given station over one calendar year) with margay detections, 57 detected more than one individual (39 location-years with 2 individuals, 11 location-years with 3 individuals, and 7 location-years with 4 individuals). We were able to identify the sex of at least two individuals for each of 37 location-years; of these, male-female dyads were most common, occurring during 25 location-years, including 8 location-years with a single female and multiple males. Dyads of two females were rare (four location-years), while dyads of two males were more common (18 location-years, including two location-years with three males).

**Dynamic interaction.**   We measured the time difference between detections of pairs of individuals in 104 dyads from the 57 location-years with multiple individuals. For approximately half (53) of the 104 dyads, the time interval between detections of different individuals was ≤ 28 days, the majority (51%) of which occurred within 7 days of each other (18% on the same day), 19% within 7 to 14 days, and 30% within 15 to 28 days. We could identify the sex of both individuals for 30 of the 53 dyads for which individuals were detected within 28 days of one another. Of the 30 dyads, detections of two consecutive females within 28 days of each other were most rare, with only two instances (7%) and the longest time intervals between detections (15 and 27 days). Eleven dyads were male-male pairs (37%), including the two

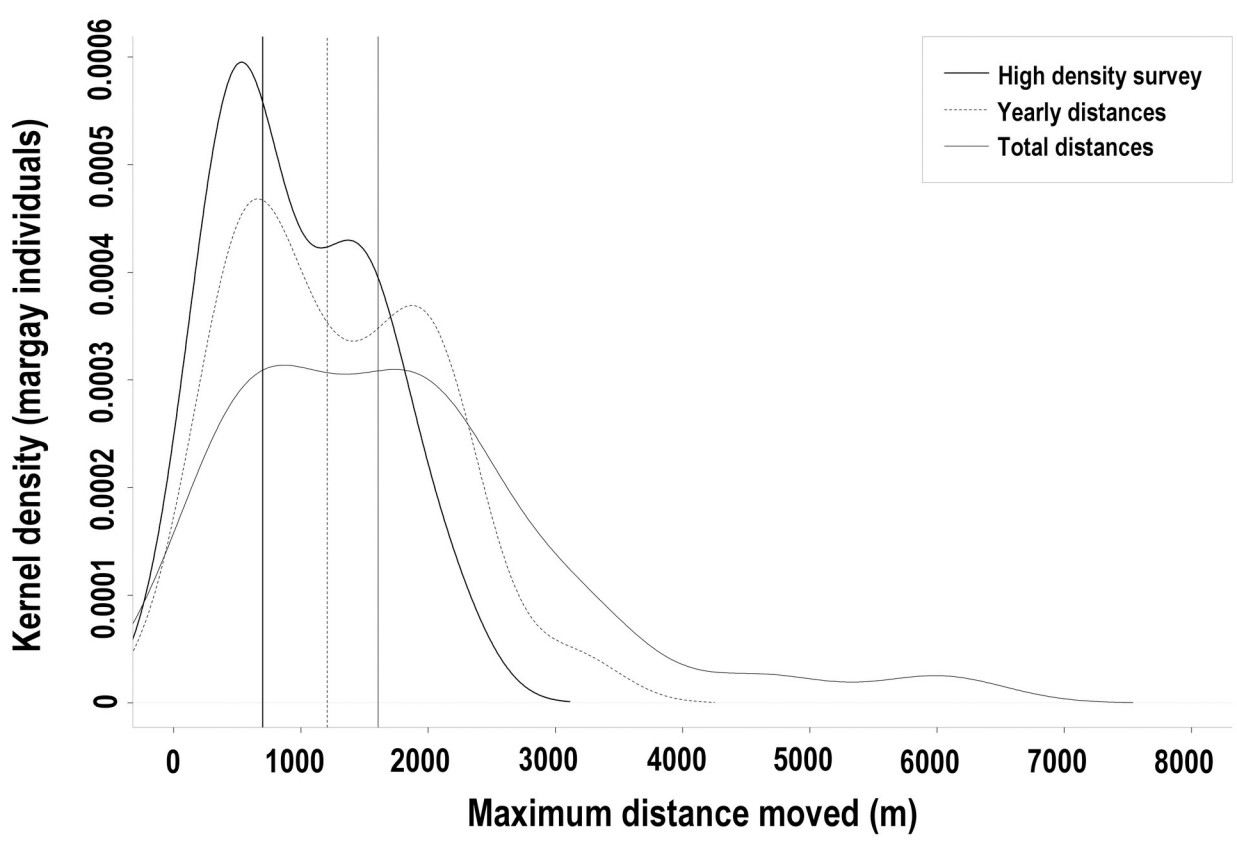

**Fig 7. Maximum detected distance between camera traps for margay individuals in the Cockscomb Basin Wildlife Sanctuary, Belize.** Kernel density distribution of the maximum distance moved (MDM) between camera stations by margay individuals detected at >1 location in the Cockscomb Basin Wildlife Sanctuary (default kernel density function with bandwidth 328.7), using three datasets differing in temporal and spatial trap effort (vertical lines show the median for each dataset: **(a)** MDM over one year, high density survey, May-2016 to Jun-2017, N = 55 stations along 14 km of trail, maximum distance between stations = 5.5 km, n = 13 individuals detected at > 1 station. **(b)** MDM per year, all stations 2003 to 2017, not allowing moves between years, N = 250 stations, maximum distance between stations per year = 21.6 km, n = 25 individuals detected at > 1 station **(c)** MDM from 2003 to 2017, all stations 2003 to 2017, allowing moves between years, N = 250 stations, maximum distance = 21.6 km, n = 32 individuals detected at >1 station.

shortest time intervals between consecutive detections (1:20 min and 5:32 min), while male-female dyads were the most common with 17 cases (57%). The interval lengths between male-male detections and male-female detection dyads had a similar distribution, with the average interval between detection ranging from 11 to 15 days (Fig 8).

## Spatial distribution of margay detections

Taking into account an effort threshold of at least 100 trap-nights per station necessary to detect a margay, we found no evidence that margay detections were clustered or patchy within the study area (Fig 9). We detected margays at 19 of the 20 annual survey stations, although not during every survey. The annual survey station where we did not detect margays was located on the access road to the CBWS headquarters. This drivable dirt road, bordered on both sides by forest, is approximately 5 m wide, with drainage ditches on either side and frequented by vehicular traffic bringing tourists and rangers in and out of the sanctuary. We detected no margays at this location despite a total trap effort of 1,879 trap-nights from 2003 to 2017.

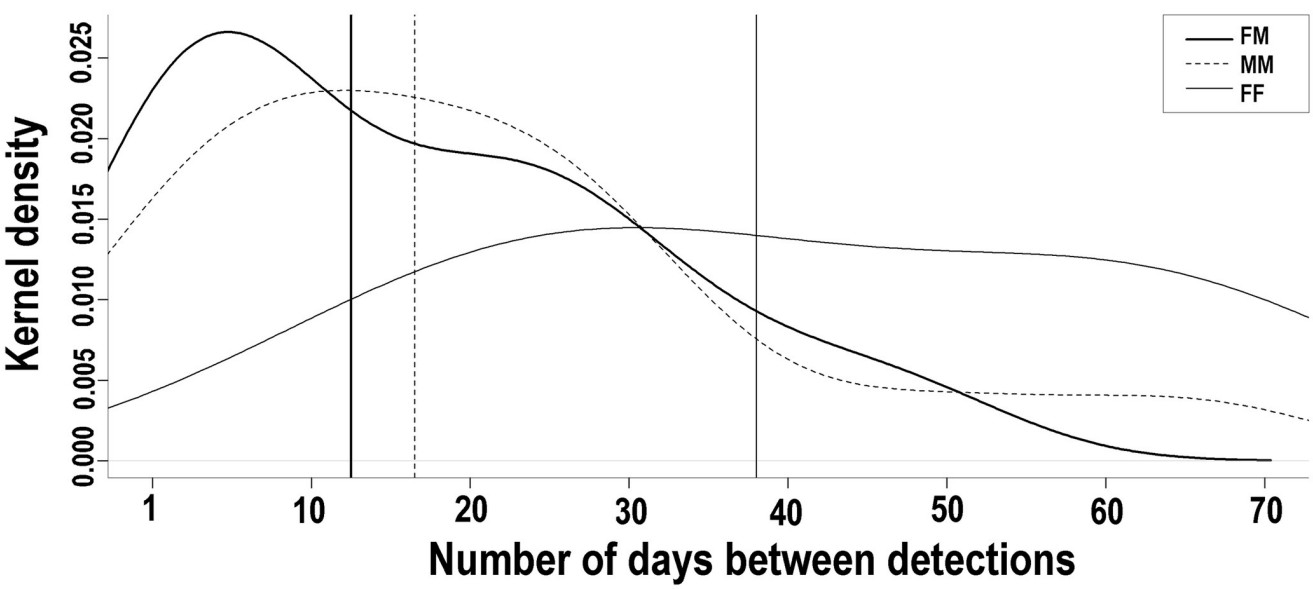

**Fig 8. Time interval distribution between consecutive captures of the different margays at the same location, in the Cockscomb Basin Wildlife Sanctuary, Belize.** Kernel density distribution of time intervals (in days) between consecutive detections of two different margays at the same camera location (dyads); showing Female-Male dyads (FM = thick line), Male-Male dyads (MM = dashed line), and Female-Female dyads (FF = thin line).

## Survival

Only 20% (37/187) of individuals were recaptured over multiple survey years. We detected half (18/37) of the individuals present in multiple survey years across two consecutive years, and only three individuals across more than four survey years (Fig 10). Additionally, 18 of the 37 individuals recaptured over multiple years were only detected at a single location, including the two individuals with longest periods between first and last detection, spanning 7 and 9 years respectively (Fig 10). The individual with detections spanning nine years was a male first detected in 2008 and recaptured in 2017. We assume that he was at least 10 years old on the date of last detection. He was detected nine times at the same location across 6 survey years suggesting consistent and stable home range use across years.

## Discussion

Recent camera trap studies suggest that margays exist at densities ranging from 0.10 to 2.4 individuals / km$^2$ [12, 25–27]. Previously, margays have been considered rare to uncommon and classified as Near Threatened on the IUCN Red List (0.01 to 0.05 / km$^2$ [10]). Species have been considered rare if their detection rates are < 0.3 detections per 100 trap-nights [69]. On this basis, should margays be considered regionally rare? Of 22 published camera trap studies that have documented margays, detection rates range by a factor of 66, from 0.04 to 2.64 detections per 100 trap-nights (mean = 0.6, S1 Data). To what extent does this reflect variability in population size versus inadequacies in the sampling method? Our results from ~ 55,000 trap-nights, spanning 14 years, sampling 236 locations across 120 km$^2$ of protected forest in the Cockscomb Basin of Belize, show that the detection of margays by trail-based cameras is low, erratic and not reliable for monitoring this species; however their spatial distribution suggests that they are not rare in this area.

While we detected a large number of individuals throughout the 14-year period, the majority were detected only once. Recaptures of individuals were sporadic, with several years

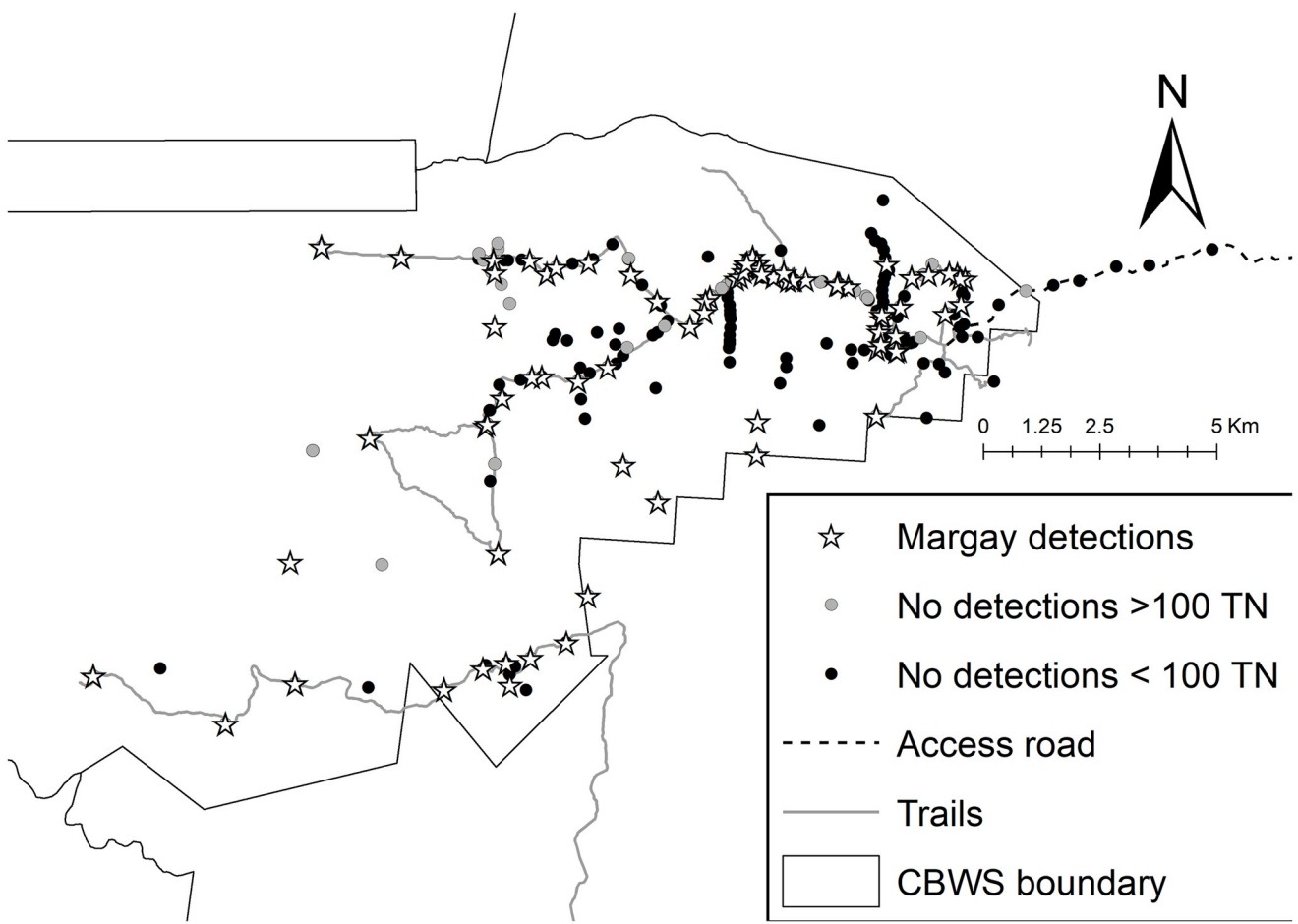

**Fig 9. The spatial distribution of margay detections on camera in the Cockscomb Basin Wildlife Sanctuary, Belize.** Black dots indicate locations of camera traps, white stars indicate margay detections (250 camera locations).

between repeat detections of long-term residents, even at the same location. Our records of detections of the same individuals over multiple years present the first estimate of margay longevity in the wild, of at least 10 years old. Spatial recaptures were rare, especially for the surveys designed for large cats (>1.5 km between stations), but increased with camera density (shorter trap spacing). We are aware of only two other camera trap studies which report on the recapture rates of margays: in the Atlantic forest, Brazil, the individual encounter frequencies ranged from 1 to 6 detections at the same location over 60 sampling days [12]; and in forest of the eastern slopes of the Andes in Ecuador, nine out of 10 margays were recaptured over 19 months, with an average of 7.5 detections per individual (range = 1 to 17), at an average of 4.5 locations per individual (range = 1 to 9 locations, with cameras spaced 200–500 m apart [14], Vanderhoff pers comm).

Although we detected margay movements of up to 6 km in our study area over the 14 years, generally moves were less than 2 km per year. As noted elsewhere, males moved further than females (0.8 versus 0.5 km, this study; 1.2 versus 0.6 km, Atlantic forest, Brazil [12]). Excluding outliers suggestive of dispersal events or exploratory moves, we estimate margay ranges in the Cockscomb Basin to be approximately 1 to 2 km$^2$. The few published studies of margay movement patterns report wide variation in home range size (1.2 to 6.0 km$^2$, mean = 4.1 km$^2$, N = 4 males, in cloud and tropical forest of northeast Mexico [70]; and 12.6 km$^2$ for one female in

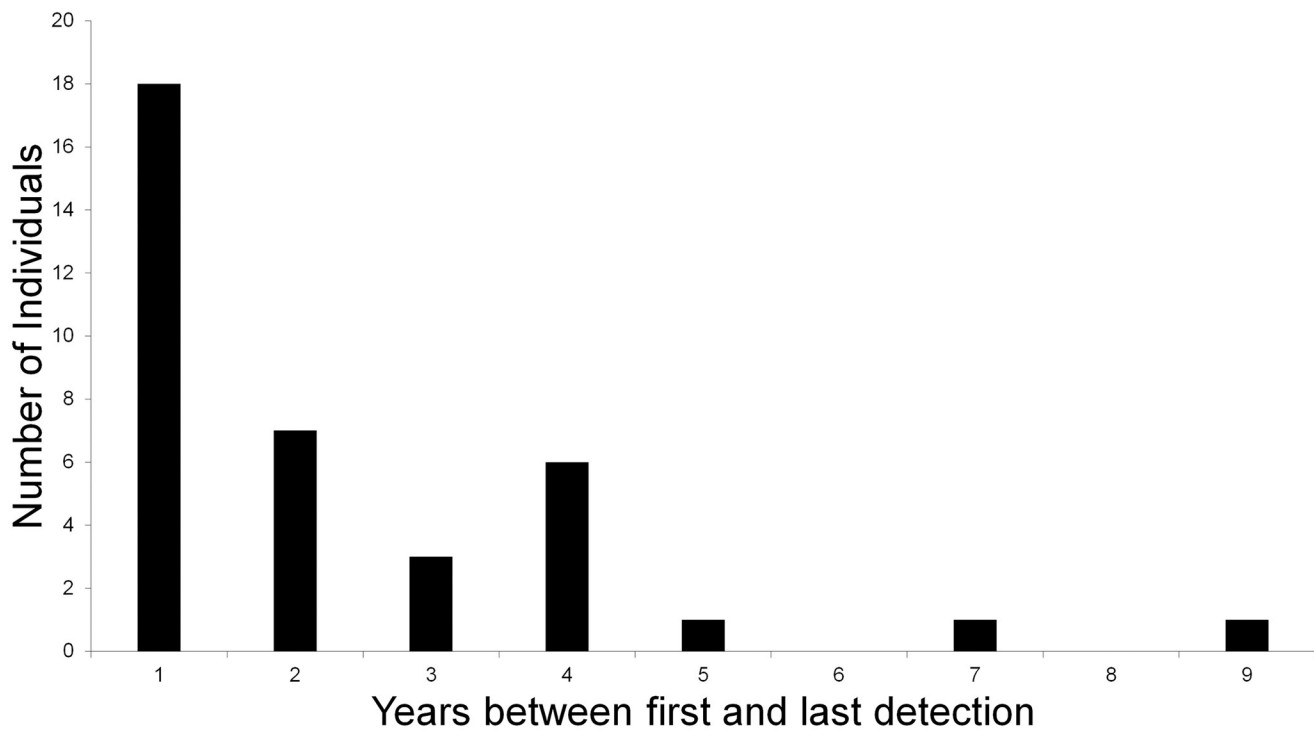

**Fig 10. Time (in years) between first and last detection of margay individuals in the Cockscomb Basin Wildlife Sanctuary, Belize.** Frequency distribution of number of individuals with varying time intervals between first and last capture detection across the total set of surveys between 2003 and 2017.

fragmented Atlantic forest and agricultural lands of southern Brazil [71]). Our crude estimates of margay ranges are small compared with these sites, and of semi-arboreal tropical cats in general [11]. Given these range sizes, we suggest that detection rates might be improved if trap spacing does not exceed 1 km (e.g. [12, 26, 30]).

Margay individuals detected across multiple years were generally recaptured at the same locations, indicating stability of home ranges or territories. Despite relatively small ranges, we found evidence of overlap and dynamic interaction both within and between the sexes, suggestive of a high-density population. Similarly [70], documented a high degree of home range overlap between males (average 30% overlap), and small exclusive core areas. Overall, our camera data suggest that the margays of the Cockscomb Basin have small, overlapping ranges, which are distributed continuously across the landscape, indicative of an abundant rather than rare carnivore. This may be expected, as the protected forest of the Cockscomb Basin is considered pristine, free of anthropogenic pressures and having year-round water availability thanks to its dense network of rivers and streams [8].

The activity of margays on the forest trails was primarily nocturnal and similar between the sexes, overlapping with jaguars, pumas and ocelots, which co-exist at relatively high densities in the area [8, 19, 50–52, 72]. It has been suggested that margay populations of the lowland tropics are impacted by intraguild predation by ocelots, with naturally low margay densities where ocelots exceed 10 individuals per 100 km$^2$ (the 'ocelot effect') [73]. Although the ocelot density in the Cockscomb Basin lies within the suggested range for such an effect (from 6.7 to 22 ocelots / km$^2$ [51]), our data indicate that margays are common. Likewise, extensive long-term diet studies in the area show no evidence of margay predation by jaguars or pumas

[74–76]. Potentially the dense structure of the Cockscomb forest and the arboreal habit of margays facilitates their avoidance of direct interaction with the larger cat species.

Our multi-season model of margays within a grid designed for monitoring jaguars (20 trail-based camera stations) shows widely fluctuating margay occupancy between survey years. The interpretation of occupancy within the context of camera traps is problematic, particularly if camera locations are not optimized for the target species. How does detection or non-detection in the ~ 30 m$^2$ field of view of the camera translate to occupancy and its extrapolation across the survey grid or landscape? We propose that the temporal variation in occupancy of the camera grid reflects the sporadic presence of margays at the point locations within the narrow field of vision of camera traps, rather than population processes. We consider as the most parsimonious explanation that the camera traps were not placed in locations routinely frequented by margays. Indeed, whether or not we detected margays depended on the length of the sampling period, not the location. We expected this, given the homogenous structure of the Cockscomb jungle. The only exception was along the entrance road to the reserve, which is approximately 8 m wide, with low grass verges on either side, and frequented by vehicular traffic for tourism. Although wide dirt roads are attractive travel routes for large cats like jaguars [4], we detected no margays here. Indeed, margays are more likely to be detected on narrow trails (0.5–1.5 m) than wide trails (> 1.5m) or dirt roads, and prefer to walk under vegetation cover than in the open [12, 30, 36]. In our study, the trails were on average 1.7 m wide [4]. The low and erratic detection rates of margays by our trail-based cameras suggest that we did not sample the regular travel routes of margays, and that the movements of margays along the trail system were rare events. Indeed, our camera traps more often detected margays crossing rather than following the trail (Harmsen pers obs). However, the detection rate increased when we increased the density of trail cameras, effectively forming a 'net' of detectors (see also [43], for which we note that the detection rate of margays increased by a factor of 4 when the trap density was doubled).

The sporadic detections of margays, the increase in detections of individuals with trap effort, and the relatively small and overlapping ranges suggest that the low and erratic detection of margays in this study area can be explained by sub-optimal sampling by trail-based camera grids designed for larger target species, rather than by the rarity of the species. Our conclusion is drawn from long-term patterns of margay detections over 14 years. We note that across the years, detection probability was constant within the occupancy framework, but detection rates ranged by a factor 10 from 0.14 to 1.47 per 100 trap-nights for the 12 3-month, 20-stationsurveys despite sampling a stable environment at permanent camera locations. This highlights the problem of using by-catch data for population assessment. Single 3-month 'snap-shot' surveys are difficult to interpret, and yet they are common in the published literature [3, 19]. Our study has shown the benefit of long-term sampling (repeated surveys), in order to make robust inferences about the study population, and we caution against the use of snap-shot data for assessing population status. Where long-term studies are not feasible, we recommend that the survey period be extended for as long as possible, taking into account the temporal limits of a closed population (e.g. [19]). Additionally, we recommend combining of margay by-catch data from the multiple camera trap efforts across the region (e.g. [12]), allowing researchers to boost sample size, explore habitat covariates and address questions of natural history and ecology at a larger scale.

Given the difficulty of detecting margays with camera traps, we expect that population assessments will underestimate their true abundance if based on short-term surveys that are not specifically designed for margays. The implications are far-reaching if applied to range-wide assessments. For example, the margay is listed as Near Threatened on the IUCN Red List, based on in particular low detection rates by camera traps, high rates of deforestation and

intraguild competition across their range [10]. [10] have proposed that competition with oce-lots may prevent margays from attaining an effective population size for long term persistence in any Conservation Unit outside the Amazon basin, although there are few data to support this. The question stands whether the range-wide inference about the population status of margays, based on low detection rates, is valid; or whether the assessment would change with the implementation of improved sampling methods for margays.

We have shown that trail-based camera grids designed for monitoring jaguars are sufficient to detect the presence of margays, and if implemented over the long-term can be used to draw inference about the population. However, by-catch data from short-term surveys are unlikely to provide an adequate sample for estimating population size, which requires repeat detections of multiple individuals (e.g. [48, 77]). Robust population assessment requires that the camera survey design is optimized for the species of interest. When deciding where to deploy camera traps, field researchers often rely on natural funnels to ensure that the target species, if locally present, will pass in front of the camera trap (e.g. along a trail or stream through dense vegetation [3, 4]). Wide trails are not optimal locations for detecting species, such as the margay, which prefer cover when walking on the forest floor (e.g. this study; [12, 30, 36]. Some researchers have recommended sampling 'off-trail' to improve the detectability of cryptic forest carnivores (e.g. [78]). However, unless located along a river or stream, a trail must be followed or created in order to deploy and maintain a camera trap in a dense forest. Furthermore, camera traps require an open area in front of the lens to detect motion. Recently researchers have started exploring the utility of deploying camera traps in the canopy for monitoring semi-arboreal carnivores [79]. However, working in the canopy is associated with additional logistical limitations and high variability in detectability between camera locations due to the structural diversity of the canopy, and as such is more suitable for species inventories and distribution than estimation of population size [79]. While it may not be useful or possible to deploy cameras 'off-trail', or in the canopy, sampling narrow trails rather than wide trails or roads will likely improve the detection rate. Increasing the density of cameras will also improve detection rates, and in the absence of additional data we would recommend camera spacing of $\leq$ 1 km.

Margays are one example of many semi-arboreal tropical carnivores living throughout the tropics of Latin America, Africa and Asia, with unknown density and abundance distributions [7]. For all these species, camera traps are the most practical means to sample their activity on the forest floor. However, we expect that the frequency and time spent ground-walking on trails, and thus detectability, will vary with the terrain, forest type and structure, width and cover of the trails, prey base and availability, and the presence of other carnivore species (predators and competitors). Consequently, detection rates of semi-arboreal carnivores by ground-based camera traps will vary within and between species, regardless of population size. Researchers have yet to formally investigate the relationship between the true abundance of semi-arboreal species and their detection frequency on the ground, and on or off trails. Doing so will allow a more meaningful interpretation of the variation in detection rates within species between sites, and between sympatric species. We hope that this first long-term study will encourage researchers to consider methodological sampling issues when assessing margay populations or other carnivores with limited information, and so distinguish between rarity and low detectability when assessing population status.

## Supporting information

**S1 Data. Detection rates of margays from camera trap surveys; extracted or derived from publshed literature.**
(XLSX)

**S2 Data.**
(CSV)

**S3 Data.**
(XLSX)

## Acknowledgments

The Government of Belize, Forest Department and Belize Audubon Society provided invaluable logistical support. We thank all Belize Audubon Society staff, both field and office, with particular reference to Nicacio Coc and Dominique Lizama for all there help over the years. For additional field assistance, we thank Emiliano Pop, Emma Sanchez, Rebecca Wooldridge, Claudia Wultsch, Miranda Davis, Marvin Vasquez, Said Gutierrez, and Paul Higginbottom. We further thank Dr. Howard Quigley, Prof. C. Patrick. Doncaster, Prof. Marcella Kelly, Dr Scott Silver and Dr Linde Ostro for their continued support over the years. We thank two anonymous reviewers, and Hugo Mantilla-Meluk for their valuable suggestions, improving the manuscript. We continue to indicate our gratitude to the late Dr Alan Rabinowitz without whom our work in the Cockscomb Basin would not be possible.

## Author Contributions

**Conceptualization:** Bart J. Harmsen, Nicola Saville, Rebecca J. Foster.

**Data curation:** Bart J. Harmsen, Nicola Saville, Rebecca J. Foster.

**Formal analysis:** Bart J. Harmsen, Nicola Saville, Rebecca J. Foster.

**Funding acquisition:** Bart J. Harmsen, Rebecca J. Foster.

**Investigation:** Bart J. Harmsen, Nicola Saville.

**Methodology:** Bart J. Harmsen, Rebecca J. Foster.

**Project administration:** Bart J. Harmsen, Rebecca J. Foster.

**Supervision:** Bart J. Harmsen.

**Visualization:** Bart J. Harmsen, Rebecca J. Foster.

**Writing – original draft:** Bart J. Harmsen.

**Writing – review & editing:** Bart J. Harmsen, Nicola Saville, Rebecca J. Foster.

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
