## [Decision Letter · Decision Letter 0]

26 Nov 2020

PONE-D-20-34073

Long-term monitoring of margays (Leopardus wiedii): implications for understanding low detection rates

PLOS ONE

Dear Dr. Harmsen,

Thank you for submitting your manuscript to PLOS ONE. After careful consideration, we feel that it has merit but does not fully meet PLOS ONE’s publication criteria as it currently stands. Therefore, we invite you to submit a revised version of the manuscript that addresses the points raised during the review process.

We look forward to receiving your revised manuscript.

Kind regards,

Bi-Song Yue, Ph.D

Academic Editor

PLOS ONE

2. In your Methods section, please provide additional location information of the study site, including geographic coordinates for the data set if available.

"This research was funded by Panthera, 590 the Wildlife Conservation Society, the UK Natural

Environment Research Council, the Liz Claiborne Art Ortenberg Foundation, the North of

England Zoological Society, Brevard Zoo, Woodland Park Zoo, Virginia Tech, and the

Summerlee Foundation. The Government of Belize, Forest Department and Belize Audubon

Society provided invaluable logistical support."

"No: The funders had no role in study design, data collection and analysis, decision to publish, or preparation of the manuscript."

6. We note that Figure 1 in your submission contain map images which may be copyrighted. All PLOS content is published under the Creative Commons Attribution License (CC BY 4.0), which means that the manuscript, images, and Supporting Information files will be freely available online, and any third party is permitted to access, download, copy, distribute, and use these materials in any way, even commercially, with proper attribution. For these reasons, we cannot publish previously copyrighted maps or satellite images created using proprietary data, such as Google software (Google Maps, Street View, and Earth). For more information, see our copyright guidelines: http://journals.plos.org/plosone/s/licenses-and-copyright.

6.1.    You may seek permission from the original copyright holder of Figure 1 to publish the content specifically under the CC BY 4.0 license. 

6.2.    If you are unable to obtain permission from the original copyright holder to publish these figures under the CC BY 4.0 license or if the copyright holder’s requirements are incompatible with the CC BY 4.0 license, please either i) remove the figure or ii) supply a replacement figure that complies with the CC BY 4.0 license. Please check copyright information on all replacement figures and update the figure caption with source information. If applicable, please specify in the figure caption text when a figure is similar but not identical to the original image and is therefore for illustrative purposes only.

Reviewers' comments:

Reviewer's Responses to Questions

**Comments to the Author**

1. Is the manuscript technically sound, and do the data support the conclusions?

Reviewer #1: No

Reviewer #2: Yes

Reviewer #3: Yes

2. Has the statistical analysis been performed appropriately and rigorously? 

Reviewer #1: No

Reviewer #2: I Don't Know

Reviewer #3: Yes

3. Have the authors made all data underlying the findings in their manuscript fully available?

Reviewer #1: Yes

Reviewer #2: Yes

Reviewer #3: Yes

4. Is the manuscript presented in an intelligible fashion and written in standard English?

Reviewer #1: Yes

Reviewer #2: Yes

Reviewer #3: Yes

5. Review Comments to the Author

Reviewer #1: The authors studied the relative abundance of margays (Felis wiedii) along a 12-years period in a natural area in Belize. Their sampling design was established according to territory of big cats (jaguar and puma), by placing 20 stations with two camera-traps each and sampled for three consecutive months during the dry season. Authors claimed about the inconvenience of sampling small cats, like margays, with large grids with long interdistances between stations. In order to understand whether perceived populations can be affected by the sampling design, they included a line transect of camera-traps with low interdistance during two years and calculated some parameters to see their influence on estimated margay abundance.

I acknowledge the huge effort performed by authors, and the data presented could be remarkably interesting for the knowledge of a very rare species like the margay. However, I see several shortcomings on the way data were analyzed, by using simple linear correlations, T-tests, Chi2 tests. In my opinion, it is not a correct statistical approach. Several claims about detectability are raised along the paper, but authors do nothing to control for it. I guess that occupancy models will help to understand whether the sampling grid used for large cats can result in low detectability for margays, and this can correct naïve occupancy to actual occupancy. Moreover, the use of GLMMs would be necessary to account for several important thinks that are wrongly analyzed, like the influence of sampling interdistance on margays relative abundance. A single model including all relevant predictors (year, sex, competitors, etc), the interdistance as a covariate, and sampling station and individual as random factors, will yield more powerful information on the data. This incredible long sampling scheme will be adequate to capture-recapture models for demography of the margays.

Furthermore, I think that the text is very difficult to read fluidly, and several paragraphs showed statements that are very subjective and not supported by any reference.

I guess that the article needs to be re-analyzed by using more adequate statistical approach.

Line 41: What means micro-placement? Please clarify

Line 43: there will be also few captures

Line 44: What means micro-level? Please clarify

Line 154: What means pre, post, please be clearer. It is problematic to gather stations with different interdistance (2000, 700, 140 m) to the original design. During the last three years interdistance was a third of the original design. Justify

Line 209. If you are not confident about the role that the capture probability has on the data, why do not applying occupancy models accounting for imperfect detectability? I guess that your data are adequate for this approach by using the 20-stations design.

Line 218. This can be true for widely spaced stations, but what happens when stations are too close like in the last three years?

Line 220. This paragraph is confuse. Try to use the two concepts (spatial vs temporal) recaptures along the paper. I guess that recaptures are a combination of both spatial and temporal recaptures.

Line 236: It is very difficult to follow this complex way of using your dataset, by adding/deleting data at your convenience.

Line 277: Occupancy models will yield information about survival, mortality, etc.

Line 284: “Most detections on the forest floor occurred at night”, but, have you detected the species at other forest levels? If not, you can delete it.

Line 285; Is this overlap high? Please, use some kind of reference to the values showed.

Line 322: Using simple linear correlations for such a data is a poor statistical approach. There are several uncontrolled variables that, maybe, can be interfering with the observed pattern.

The statistical approach used is rather dull, when there are very flexible and powerful tools easy available like GLMMs that would be suitable for such a heterogeneous data set. Using the number of independent detections by individual/year as the response variable, you can model several responses to grid design, year, sex, competitors, recaptures, and so on, while controlling for the spatial situation of stations (random effects), the identity of individuals (random effects), and the sampling effort. In a similar way, actual occupancy can be examined controlling for imperfect detection by application of occupancy models and assessing the probabilities of local colonization and extinction along the study period. Individuals’ capture histories surely will yield some interesting data about survival and other demographic parameters. And even, interesting information about density.

There are important reasons for rarity of margays in the study area: the competence with ocelots, and the arboreality, which means less detectability in stations placed on the forest floor.

Reviewer #2: This manuscript reflects an exceptional effort to improve our knowledge on a species that is difficult to study. It is commendable to see a publication of a very long study effort, which showed the benefit of long-term research on a seemingly rare animal. Apart from several minor grammatical pick-ups (please see below), I would only have a few suggestions that may improve this publication.

1) The authors should reflect more on the ecology and life history of margays. What is known about their habitat selection (e.g. is this study site thought to be a representative habitat of where these cats would thrive, given the background of the site – Line 129. This of course would have implications on the density of these animals in the area)? Similarly, what is the life history of margays – expected longevity (is there anything known about captive populations for instance, or other similar felids), reproductive output; and how does this link with the results obtained?

2) How does the high density of other felids in the area (pumas, ocelots, and jaguars) (Line 135) impact on margays? Is there interspecific competition for prey, predator- prey interactions, other interspecific competition, that is expected to impact on margays? If the author’s expectation is different to the observation, please provide some suggestions of why you may not have observed the expected outcomes? The manuscript briefly touched on this in the discussion, but the “ocelot effect” was not observed (despite the high ocelot density), and the explanation for this was not explored/provided. Similarly, for instance, smaller home ranges observed (Lines 510-513) could be a reflection of inter- and intra- specific competition, rather than just camera placement?

3) Finally a small improvement on the conclusions could be to reflect on suggested optimal time the cameras should be deployed for (i.e. what does a ‘long-term study’ mean – would 2-3 years be sufficient? 12 years is possibly an unrealistic expectation, and/or other improvements should be included). If these cats are semi-arboreal, is there a chance of changing the placement of the cameras and using them in canopy (at what stratification would you expect margays?). In addition, can scat analyses, tracks and other markings be of use?

4) The authors have suggested that Animal Ethics approval was not applicable to this study? Why was this not required for a camera trap study?

Lines 50-52 – examples are convoluted – consider revising sentence.

Line 86 – parenthesis “(“ can be deleted.

Line 103 – missing full-stop after the word design.

Line 139 vs 142 state that 19 vs 20 locations were surveyed for 12 seasons

Line 154 put a space after pre-,

Lines 168 -171 are meant to go above the table (same for other table captions)

Line 207 locations-years should be location-years

Line 237 and 240 and 242 put space before km (check throughout the manuscript)

Line 330 need a space before .

Line 333 need a space after 0.0044

Line 336 need a space after 0.53

Line 156, 452, 527, 528 need space between number and m (check throughout the manuscript)

Line 480 need a parenthesis “)” after [13]

Line 494 clarify why/how your results are suggestive of a stable population

Line 502 missing full stop

Line 510 missing parenthesis “)”

Line 513, 534. 543, 553 references are spelled out

Line 540 14 years or 12 years?

Line 566 change [] to ()

Reviewer #3: The manuscript is well organized and compiles 12 years of research in detailed analyses, regarding detectability. I made few comments on some parts of the text, listed at the end of this concept.

I just have a couple of comments regarding the discussion section. As mentioned by the authors, the study site at Cockscomb Basin Wildlife Sanctuary, represents a pristine area with overall good conservation conditions. To what extend it is safe to extrapolate data from a pristine area like that, to other study sites likely to hold a different quality of habitats for margays? This is of particular interest, since some instances of the inferences made by the authors refer to the conservation status assigned by the IUCN to the species.

A paragraph talking about the implications of habitat quality on felid species behavior and its effect on results obtained from camara trap assessments would be great.

On the other hand, there is little discussion on the differences in number of detections throughout the years at the study site. I am pretty sure, that based on their knowledge of the study area, the authors will be able to add some information on other environmental conditions that could potentially drive this changes in detections (i. e. climate variability, specially, El Niño event that was particularly strong in 2015-2016, and La Niña event, that reported a strong effect between 2010-2011, these two lapses yielded low values).

It would be of great help for the readers to know about the thoughts of the authors on these issues.

Taking into consideration the quality and robustness of the information presented, I consider the manuscript to be published with minor additions and changes.

SUGGESTIONS

Lines 56 and 57: please review closing parenthesis, starting at: “(e.g. jaguars (Panthera onca): …”.

Lines 64 and 65: please reorganize the sentence: “we assess population status of a neotropical, semi-arboreal felid, the margay (Leopardus wiedii), …” Suggestion: we assess population status of the margay (Leopardus wiedii),a neotropical, semi-arboreal felid,…

Line 68: include “As mentioned” in: Margays are small, semi-arboreal felids weighing 2 to 4 kg,…”; suggestion: As mentioned, margays are small, semi-arboreal felids weighing 2 to 4 kg,…

Lines 70 to 72: sounds a little contradictory, that the authors affirm: “The geographic ranges of both species almost completely overlap, making margays potential by-catch in camera trap survey grids designed for jaguars [13,14].” which is just the point of controversy of this particular study. It would be more coherent, in the context of the logic of the paper, for the authors to previously introduce a sentence with the following statement: It has been assumed that species that overlap in their geographic ranges and are considered sympatric in some areas can be effectively detected under single experimental designs, that not necessarily take into consideration behavioral microhabitat preferences.

Lines 92 to 94: We address this by presenting a simple method to assess rarity based on the spatial characteristics of a population and trapping effort, and apply it to the margay ‘by-catch’ data obtained from a long-term jaguar-monitoring program. The authors use the same heading of this sentence in line 111: “Here, we use a simple method to make inferences about the population status of margays…” it would sounds better if the authors say: Here we used the above mentioned approach to make inferences about the population status of margays…

Lines 129 to 130: It is important to provide a geographic context associated with the study site, suggestion: “The study was conducted in the moist broad-leaved tropical forest of the Cockscomb Basin Wildlife Sanctuary (from hereon, CBWS or sanctuary), a natural area located in the south-central zone of Belize created to protect the forests, the fauna and the hydrographic basins of approximately 400 square kilometers of the eastern slopes of the Maya Mountains…

Line 502: include a period at the end in: “cameras spaced 200 – 500m apart [12], Vanderhoff pers comm).

6. PLOS authors have the option to publish the peer review history of their article (what does this mean?). If published, this will include your full peer review and any attached files.

Reviewer #1: No

Reviewer #2: No

Reviewer #3: **Yes: **Hugo Mantilla-Meluk

---

## [Author Response · Author response to Decision Letter 0]

3 Feb 2021

For reviewer comments, see cover letter

---

## [Editor Report · Decision Letter 1]

9 Feb 2021

Long-term monitoring of margays (Leopardus wiedii): implications for understanding low detection rates

PONE-D-20-34073R1

Dear Dr. Harmsen,

We’re pleased to inform you that your manuscript has been judged scientifically suitable for publication and will be formally accepted for publication once it meets all outstanding technical requirements.

Kind regards,

Bi-Song Yue, Ph.D

Academic Editor

PLOS ONE

---

## [Editor Report · Acceptance letter]

16 Feb 2021

PONE-D-20-34073R1 

Long-term monitoring of margays (*Leopardus wiedii*): implications for understanding low detection rates 

Dear Dr. Harmsen:

I'm pleased to inform you that your manuscript has been deemed suitable for publication in PLOS ONE. Congratulations! Your manuscript is now with our production department. 

Kind regards, 

on behalf of

Dr. Bi-Song Yue 

Academic Editor

PLOS ONE